# Weekly External Load Performance Effects on Sports Injuries of Male Professional Football Players

**DOI:** 10.3390/ijerph20021121

**Published:** 2023-01-08

**Authors:** Francisco Martins, Adilson Marques, Cíntia França, Hugo Sarmento, Ricardo Henriques, Andreas Ihle, Marcelo de Maio Nascimento, Carolina Saldanha, Krzysztof Przednowek, Élvio Rúbio Gouveia

**Affiliations:** 1Department of Physical Education and Sport, University of Madeira, 9020-105 Funchal, Portugal; 2LARSYS, Interactive Technologies Institute, 9020-105 Funchal, Portugal; 3CIPER, Faculty of Human Kinetics, University of Lisbon, 1495-751 Lisbon, Portugal; 4ISAMB, Faculty of Medicine, University of Lisbon, 1649-020 Lisbon, Portugal; 5Research Center in Sports Sciences, Health Sciences, and Human Development (CIDESD), 5000-801 Vila Real, Portugal; 6University of Coimbra, Research Unit for Sport and Physical Activity (CIDAF), Faculty of Sport Sciences and Physical Education, 3004-504 Coimbra, Portugal; 7Marítimo da Madeira—Futebol, SAD, 9020-208 Funchal, Portugal; 8Department of Psychology, University of Geneva, 1205 Geneva, Switzerland; 9Center for the Interdisciplinary Study of Gerontology and Vulnerability, University of Geneva, 1205 Geneva, Switzerland; 10Swiss National Centre of Competence in Research LIVES—Overcoming Vulnerability: Life Course Perspectives, 1015 Lausanne, Switzerland; 11Department of Physical Education, Federal University of Vale do São Francisco, Petrolina 56304-917, Brazil; 12Institute of Physical Culture Sciences, Medical College, University of Rzeszów, 35-959 Rzeszów, Poland

**Keywords:** soccer, risk factors of injury, injury prevention, GPS, sports monitoring

## Abstract

One of the most challenging issues professional football players face throughout their careers is injuries. Those injuries often result from suboptimal training programs that were not designed according to the players’ individual needs. This prospective study aimed to examine in detail the effects of sports injuries on professional football players’ weekly external load performances. Thirty-three male professional football players were monitored using 10-Hz Global Positioning System (GPS) units (Apex pro series, StatSports) during an entire season. The variables considered in the analysis were total distance (TD), high-speed running (HSR), accelerations (ACC), and decelerations (DEC). The comparisons were made between the four-week block before injury (–4T), four-week block after return (+4T), and players’ season averages (S). Players displayed significantly higher values of TD, HSR, ACC, and DEC in the –4T, compared to the other two moments (+4T and S). Furthermore, the comparison between the +4T and S showed no significant variations in the GPS metrics. It was shown that a significant increase in players’ weekly external load performance over a four–week period may have a negative effect on the occurrence of injuries from a professional football standpoint. Future research should consider the effects of injury severity on players’ external load variations.

## 1. Introduction

Sports injuries are one of the most challenging problems of professional football players throughout their career [1]. At all competitive levels, reducing injuries that prevent players from competing is a top priority for sports organizations [2], particularly from a professional standpoint, which involves a huge amount of money [3,4,5].

Worldwide, the topic of injury has attracted the attention of researchers, sports agents, and coaches [6]. According to previous studies, it is crucial to adopt a multidisciplinary approach, including nutrition, sleep quality, and training program characteristics, to act proactively and on time to prevent injury [2]. A prospective cohort study initiated and funded by the European Football Association, was conducted in professional football from 2001 to 2019, with the participation of 3302 players, comprising 49 teams from 19 countries [7]. The authors concluded that the overall injury rate of professional football players in training and competition had decreased significantly [7]. Those results testify to the importance that has been given to reducing the risk of sports injuries in professional football [8].

Clinically speaking, most studies conducted into soccer injury incidence reveal that the lower limbs are the body part most commonly impacted by sports injuries, especially when it comes to groin, quadriceps, and thigh muscle injuries [8,9,10,11,12,13,14,15]. In terms of injury severity in professional football, past studies conducted on professional football teams in Chile, Spain, and Brazil have reported an average of 7 to 8 absent days per injury [9,16,17].

However, analyzing such data may provide insufficient information for coaches and their technical staff to make decisions regarding designing training sessions, particularly concerning external demands. Football clubs often use Global Positioning System (GPS) data to monitor the training process [18,19]. Wearable GPS monitoring systems that track a player’s workload over time during training and matches are one of the most used technical improvements of the last few years [20]. In addition, external load metrics have been used to describe the pattern and severity of injuries [21]. As an important variable, the need to design training programs considering external demands has emerged [22,23,24]. Even though, it is crucial to consider other variables such as internal load, physiological response to demand, and readiness to retrain when structuring training programs.

In a study conducted over two seasons on 32 youth footballers of 17.3 ± 0.9 years of age, it was revealed that injury risk was greater when a high number of accelerations (ACC) were accumulated over a 3 week period and when the intensity of training sessions was also higher, and contact injury risk was greater when TD and high speed distance were very high [25]. In this study, knee and foot injuries were the most commonly observed, according to the classification of contact or non-contact injuries. However, a detailed global characterization of injuries and their relationship with the external demands that the team was exposed to over the two season was not their focus. It was more individualized and not a collective study, as ours was meant to be. This study also had a limited sample and worked on elite youth football players, being distinct from the representative population of our research.

Overall, the literature has contributed extensive data to the football world in terms of the analysis of games and training. However, it is still necessary to develop a greater awareness of how essential it is to adopt workload measures that focus on the balance between minimizing the risk of injury and preparing players for upcoming physical competition demands [21]. Our novel contribution will be in the global analysis of the effects that the sports injuries a specific football team faces throughout an entire sports season have on players’ weekly external load performance, measuring a set of external variables that directly influence the fatigue, performance, and recovery rates of professional football players. In addition, collecting daily data over a complete sports season of a professional football team increases the quality and novelty of our scientific research, particularly from the complexity of doing it in a professional context. Finally, less important data, about the relationship between weekly external load performance and the occurrence of sports injuries in the Portuguese First League, is lacking.

Therefore, the aims of this study were twofold: (i) to characterize the sports injuries of a Portuguese professional football team, and (ii) to examine in detail the relationship between weekly external load performance (TD, HSR, ACC and decelerations (DEC)) and the occurrence of sports injuries in this professional football team.

## 2. Materials and Methods

### 2.1. Study Design

This prospective study was conducted on a professional men’s football team competing in the Portuguese First League during the 2021/2022 season. The research was conducted from 28th of June 2021 to May 15th of 2022, according to the principles of the Declaration of Helsinki, and all stakeholders signed informed consent for participation in this study. All procedures applied were approved by the Ethics Committee of the Faculty of Human Kinetics (CEIFMH No. 34/2021).

### 2.2. Participants

Thirty-three male professional football players (25.9 ± 3.8 years old; 182.1 ± 6.9 cm; 74.2 ± 6.7 kg) participated in this study: four goalkeepers (12.1%), nine defenders (27.3%), ten midfielders (30.3%), and ten forwards (30.3%). Twenty-six players had the right lower limb as dominant (78.8%). All players who represented this team in the 2021/2022 season were included in this study, even those who joined it during the season.

### 2.3. Injury Report

Regarding the variables under consideration, lay-off days was suggested as the number of days until the player resumed full team training [26]. The part of the body that experienced structural and/or functional alterations is determined by the type, zone, and specific location of the injury. The number of days from the player’s stoppage until they can return to field work with the clinical department’s approval determines the injury’s severity. The classification of this variable was as follows: (1) Minimal, 1 to 3 days; (2) Mild, 4 to 7 days; (3) Moderate, 8 to 28 days; and (4) Severe, +28 days. The injury occurrence is characterized by the work session (training or competition) that the athlete was performing when the injury was contracted. The exposure time of the athletes throughout the season was collected using a 10-Hz GPS unit (Apex pro series, StatSports, Northern Ireland) during each training session and an official match. The injury incidence was calculated as the number of injuries contracted during a sporting activity divided by 1000 h of exposure time, multiplied by the exposure time collected with the GPS device in game and training situations.

The medical department kept daily injury records during the season, according to the mentioned variables. This department is run by the team doctor. This professional has vast experience in the sports area and has been with the club professionally for many years. He is responsible for making injury records in agreement with the physiotherapist and physical trainers. The goal was to have a multidisciplinary dialogue between the professionals involved in the process so that a consensus could be reached between the interpretation and classification of the sports injuries that occurred in this professional football team. The players injured at the end of a season were followed until the end of their recovery period.

### 2.4. Global Positioning System

The professional football players’ daily external load data were monitored using a 10-Hz GPS unit (Apex pro series, StatSports, Northern Ireland). Data were collected for one competitive season, spanning 46 weeks, from 28th of June 2021 to May 15th of 2022. The work time of the season and other precise metrics such as TD, ACC, and DEC were collected. Table 1 presents the GPS metrics considered in this study. A skin-tight bag containing the GPS gadget was placed in the thoracic region between the scapulae. The placement, collection, and verification of the data recorded by the GPS devices were completed daily by the team’s physical trainer. It is essential to reinforce that the calculation of the average work time for the season was completed by considering all the training sessions and official matches. That is, the average training and competition time considered the sessions in which the players participated and those in which they did not due to sports injuries. Using this methodology, it is possible to characterize the average time spent by the player in an average work week over the season.

### 2.5. Experimental Approach

The players’ weekly external load performance (work time, TD, HSR, ACC, DEC) was collected and analyzed in the four weeks before the injury occurred (–4W), and during the four weeks after the players’ return to their team’s work (+4W). It is important to reinforce that data collection regarding +4W refers to the moment when the medical department declared and authorized the player to fully return to training and competition, so that there were no constraints and impediments regarding the training and competition participation process. Additionally, both 4-week periods were compared to the season average (S) to examine the existence of significant variations. The season external load performance average was calculated considering not only the days that players participated in training sessions or official matches, but also considering the missing days due to injury or coaches’ match options. Indeed, those days also play significant roles in the players’ external demands in a sports season. Therefore, we aimed to calculate the season average value as a representative variable of the full conditions of the season, without the exception of any conditioning factors that players and coaches cannot control. Thus, the choice of these time periods is related to the fact that most of the recent studies in this area have followed these guidelines, as can be seen in a recent systematic review of professional football players on the topic of workloads and injury risk [27]. Therefore, it becomes possible to compare results and draw conclusions and reflections from what happened with scientific studies in the same area.

### 2.6. Statistical Analysis

Descriptive statistics were used to summarize the data collected as means ± standard deviations. Absolute values present the number of football players, the total number of injuries, and the total number of injured players. The frequency of injuries by zone, type, specific location, and severity are represented by absolute values and their percentages. The Wilcoxon Signed Rank Test was conducted to investigate differences between pre-injury, after-injury, and average season periods for GPS metrics (Work time, TD, ACC, and DEC). Effect size (*r*) was calculated by dividing the *z* value by the square root of the sample size [28]. All analyses were performed using IBM SPSS Statistics software 28.0 (SPSS Inc., Chicago, IL, USA). The significance level was set at *p* ≤ 0.05.

## 3. Results

In terms of injury reports, this team had 15 players who contracted 24 sports injuries across the season 2021/2022. On average, each player had 0.7 injuries. The typical number of days a player needed to recover from an injury was close to 17 days, with a total of 416 missed by the players of this professional team across the analyzed sports season. Overall, most of injuries occurred in training situations. However, the incidence of injuries was considerably higher in matches since the exposure time in training was substantially higher that in matches across the whole season. The full report is presented in Table 2.

Table 3 shows the injury zones, types, and specific locations. Over the season, the lower limbs were massively affected by injuries in both number and total days of recovery. The most frequent injuries were muscular, in the hamstrings and quadriceps. Sprains affecting the tibiotarsal and knee structures were also very common. Although these injuries occurred more frequently, their severity was mostly classified between mild (4–7 days) and moderate (8–28 days), as presented in Table 3. The sports injuries that led to the greatest absence in days from training and competition by the players on this professional football team were shoulder luxation/dislocation and injuries affecting the knee and adductor structures.

Descriptive statistics and results of Wilcoxon Signed Rank test to examine differences in the GPS metrics before and after injury are presented in Table 4. Firstly, the average time each player spent on the field decreased significantly when comparing the period before and after the injury occurred, as the players had an average of 105 min less in the 4 weeks after they were clinical discharged (*z* = −3.772, *p* = 0.001, *r* = 0.56, large effect size). In addition to this clear decrease, there was also a drop in the other metrics controlled by GPS, when comparing the pre- and post-occurrence of sports injuries, such as: TD (*z* = −3.802, *p* = 0.001, *r* = 0.56, large effect size), HSR (*z* = −4.015, *p* = 0.001, *r* = 0.59, large effect size), ACC (*z* = −3.711, *p* = 0.001, *r* = 0.55, large effect size), and DEC (*z* = −3.346, *p* = 0.001, *r* = 0.49, medium effect size). When comparing the GPS metrics in the four weeks before an injury occurred and the average of the entire sports season, we found that the pre-injury values were significantly higher in all analyzed metrics: work time (*z* = −3.772, *p* = 0.001, *r* = 0.56, large effect size), TD (*z* = −3.893, *p* = 0.001, *r* = 0.57, large effect size), HSR (*z* = −4.015, *p* = 0.001, *r* = 0.59, large effect size), ACC (*z* = −3.802, *p* = 0.001, *r* = 0.56, large effect size), and DEC (*z* = −3.741, *p* = 0.001, *r* = 0.55, large effect size). Finally, analyzing the same metrics and comparing them in the four weeks after the players were discharged and the average of the sports season, no significant results were found in any of the GPS metrics analyzed: playing time (*z* = −1.460, *p* = 0.144, *r* = 0.22 small effect size), TD (*z* = −1.125, *p* = 0.26, *r* = 0.17, small effect size), HSR (*z* = −0.730, *p* = 0.465, *r* = 0.11, small effect size), ACC (*z* = −0.791, *p* = 0.429, *r* = 0.12, small effect size), and DEC (*z* = −0.304, *p* = 0.761, *r* = 0.04 no effect size).

## 4. Discussion

This study aimed to characterize the sports injuries of a professional football team and analyze in detail the effects of significant variations in the values of external load (TD, HSR, ACC, and DEC) on the occurrence of sports injuries. Our novel contribution is in the global analysis of the effects of sports injuries on players’ weekly external load performance, measuring a set of external variables that directly influence the fatigue, performance, and recovery rates of professional football players. In this study, injuries in the lower limbs were the most frequent compared to injuries in other body’s areas. These were mostly characterized as quadricep and hamstring muscle injuries and knee and tibiotarsal sprains. In terms of the external load, it was shown that the increase in the values reached by the TD, HSR, ACC, and DEC analyzed over four weeks makes players more susceptible to contracting all kinds of sports injuries.

Of the 33 players on this professional football team, only 15 suffered sports injuries throughout the season. A total of 24 injuries were recorded, with an average of 0.7 per player. Previous research that was made across three seasons indicated a greater injury frequency (1.5 injuries per player) [9], than the one observed in our study. The variance in injuries per player and team might be related to the level of competition, type of preparation, and training environment each team offers. Depending on the situation and level of competition, the internal and external loads in training and matches are directly associated with sports injuries [15]. In the current study, sports-related injuries caused players to miss anywhere between 8 and 28 days of training and competition, with an average of 15.6 days per injured player. This result is higher than the ones in the previous literature, which pointed out an average of 7 to 8 missing days due to injury [9,16,17]. In the specific case of this team, it is possible that the average days missed due to injury was negatively inflated (i.e., more days away due to injury) due to the severities of concussion and luxation. In addition, knee injuries also showed a high severity, again influencing the average number of days missed due to sports injuries.

Meanwhile, most of the injuries reported occurred in the lower limbs, mainly through muscle injuries in the hamstrings and quadriceps, and sprains in the knee and tibiotarsal structure. Moreover, these injuries led to players having the most days off due to injury in the analyzed sports season. Indeed, previous research has consistently reported the lower limbs as the body zone most injured in football [9,10,11,16,29,30,31]. On the other hand, our analysis indicated that 48.8% of the players’ missing days were related to muscular injuries, which is higher than the results observed in the previous literature (ranging from 20% to 37%) [31,32,33,34,35]. Muscle injuries require an average of 2 to 3 weeks of recovery time, taking up a considerable amount of time and making it notorious in the percentage of days missed by players due to sports injuries. Therefore, it seems crucial to include training content designed to prevent muscle injury occurrence according to players’ characteristics.

Worldwide, GPS technologies are the main instrument used to monitor external load in football. According to the literature, TD, HSR, ACC, and DEC are predictive factors of players’ risk of injury [36].The relationship between external load and the likelihood of sports injury occurrence has recently been discussed using those GPS metrics [25,37,38,39,40,41]. Indeed, a recent systematic review on this topic reinforced the importance of monitoring players’ external load across training sessions [27]. Regarding the analyzed external load metrics, our results sustained that the players are more prone to suffer any type of sports injuries due to a significant increase in TD, HSR, ACC, and DEC, when comparing the average of a four-week block before the injury with the seasonal average and the four-week block after clinical discharge. In contrast, no significant variations were found when comparing the seasonal and four-week block averages after players were clinically discharged to return to training sessions. In practical terms, injuries seemed to appear when players were exposed to a higher workload and intensity. This conclusion raises the hypothesis that higher load and intensity in weekly sessions may be associated with a greater likelihood of professional football players suffering sports injuries. Thus, daily dialogue with the players to perceive internal load should be promoted to better understand their physical and psychological states [42,43,44]. The ideal condition would be to join daily external and internal loads as complementary measures since both can lead to different risk profiles [45].

Concerning TD, a study made among youth professional football players concluded that a higher weekly cumulative load in TD led to a higher risk of an overuse injury [46]. In addition, another study conducted on youth football stated that a significant increase in training intensity led to an increase in the risk of injury occurrence [25]. In our study, the TD in the four-week block before the injury occurred was 27% and 33% higher than the four-week block after clinical discharge and the seasonal average, respectively. This difference may not only be related to the significant increase in training time, but might also reflect the increase in external demands and its intensity in specific moments of the season. In official matches and training sessions, it is vital to emphasize TD as a valuable metric. Indeed, the literature has claimed that when assessing the physical demands of football, TD is a non-essential indicator [47,48]. However, in a study in a team competing in Croatia’s highest national football league, the authors reported that TD by professional players influenced the players’ rate of perceived exertion (RPE) response during training sessions [49].

When studying the physical demands of football, a lot of focus has been placed on distance at HSR [50,51]. Our study showed a significant increase in HSR in the four-week block before the sports injuries occurred, which is consistent with a previous investigation conducted on 32 elite youth football players [25]. Moreover, the authors reported that increased HSR values also increase the injury risk [25]. Conversely, a study among 37 professional football players highlighted that players who presented superior results in terms of HSR over 4 weeks of training were associated with a lower risk of suffering a sports injury [52]. However, the most interesting conclusion drawn by the authors was that the players most likely to suffer injuries were those who showed large weekly changes in HSR distances [52]. This data shows that well-developed lower-body strength, repeated sprint ability, and speed are associated with a better tolerance to higher workloads and reduced risk of injury in team-sport players [40].

Furthermore, the values of ACC and DEC were significantly higher in the four-week block before injury occurrence compared to the other two moments (+4T and S). These results indicate a relationship between the higher number of ACC and DEC and a higher risk of a sports injury. A study developed with 32 youth professional football players across two seasons described that the increase in the number of ACC over 3 weeks was associated with a superior overall injury occurrence [25]. According to our analysis, ACC predominance was greater than DEC actions. A recent systematic review reported the same conclusion [53]. However, this result should not be generalized. Indeed, the authors state that the different training methodologies used to promote specific technical and tactical objectives should affect the frequency of ACC and DEC actions by players [53]. In practical terms, it is maybe incoherent to limit the running loads of football players to lower the risk of injury because players may unintentionally be put at a higher risk of injury has they might not be prepared during crucial phases of competition when they are stimulated to push themselves to their physical edges [54]. Football involves a number of acyclical shifts in activity, each characterized by ACC and DEC, that further raise the energy demands placed on the players [55,56,57]. Therefore, when assessing the general physical demands of team sports such as football, ACC and DEC play a significant role in their characterization [55,56,57].

There are some limitations to this study that need to be acknowledged. Firstly, this study represents a small collective regarding the report of overall seasonal injuries (only one team and one season). Besides that, this investigation was carried out throughout the coronavirus pandemic, meaning the results may not be representative. Additionally, our analysis were focused on the relationship between injuries and general weekly external load metrics. A deep study of the relationship between different HSR thresholds, for example, and muscle injuries, maybe a good contribution to better understand under what intensities the injuries were sustained. Additionally, our analysis did not consider the possible influence of competitive moments. However, our study brings relevant insights to the football context, mainly by focusing on the relationship between sports injuries and the variation of weekly external load performances. Monitoring players’ external load on a weekly basis could be relevant to prevent injury by allowing sports agents and coaches to better understand the impact of the weekly workloads. In addition, it is recommended to use external and internal load as complementary measures to more detailed knowledge about players’ status.

## 5. Conclusions

In this novel investigation, the results portray that lower limb injuries happen on a large scale throughout a sports season, mainly characterized as muscle injuries in the quadriceps and hamstrings and sprains in the tibiotarsal structure and knees. The major finding of this investigation showed that the increase in players’ external load, TD, HSR, ACC, and DEC across a four-week block was related to a higher risk of players suffering all kinds of sports injuries in this professional team when comparing it with the four-week block after clinical discharge and the seasonal average. Based on these results, it is becoming more and more crucial for the coaching staff to structure their working weeks according to the achievement of two primary objectives: the physical and psychological preparation of the players for the upcoming competitive challenges and the balance in the load and intensity of the training sessions, ideally individually tailored according to the physical needs of each player. If these assumptions are not fulfilled, professional players will be at higher risk of suffering from sports injuries and, consequently, will not be able to contribute in official matches.

## Figures and Tables

**Table 1 ijerph-20-01121-t001:** GPS metrics measured in this study, their united, and zones.

Variable	Unit	Zone
Work time	Minutes	
Total distance (TD)	Meters	
High-speed running (HSR)		Above 5 m/s
Accelerations (ACC)		Under 3 m/s
Decelerations (DEC)		

**Table 2 ijerph-20-01121-t002:** Injury report during 2021/2022 season.

Players of this Team	33
Injured players	15
Total injuries	24
Total lay-off days due to injuries	416
Average lay-off days per injury	17.3
Average number of injuries per player	0.7
Injury Occurrence	
Total	24
Training	17
Match	7
Injury Exposure (h)	
Total	8430
Training	7780
Match	650
Injury Incidence *	
Average	2.9
Training	2.2
Match	10.8

* Injury incidence (per 1000 h), h (hours).

**Table 3 ijerph-20-01121-t003:** Injury characterization per zone, type and specific location during 2021/2022 season.

	No. (%)	Missed Days	Average Missed Days	Severity *
Zone				
Head	0 (0%)	0	0	none
Upper limbs	2 (8.3%)	87	43.5	severe
Trunk	1 (4.2%)	6	6	mild
Lower limbs	20 (83.3%)	319	15.9	moderate
Other	1 (4.2%)	5	5	mild
Type				
Concussion	1 (4.2%)	57	57	severe
Luxation/Dislocation	1 (4.2%)	87	87	severe
Sprain	8 (33.3%)	60	7.5	moderate
Muscle	11 (45.8%)	206	18.7	moderate
Bone	2 (8.3%)	7	3.5	mild
Other	1 (4.2%)	5	5	mild
Specific Location				
Shoulder	2 (8.3%)	87	43.5	severe
Abdomen	1 (4.2%)	6	6	mild
Adductor	2 (8.3%)	76	38	severe
Hamstring	3 (12.5%)	55	18.3	moderate
Quadricep	2 (8.3%)	15	7.5	mild
Knee	5 (20.85%)	125	31.3	severe
Calf	2 (8.3%)	28	14	moderate
Tibiotarsal	5 (20.85%)	15	3	minimal
Foot	1 (4.2%)	2	2	minimal
Other	1 (4.2%)	5	5	mild

* Average days missed by players due to a sports injury; minimal (1–3 days), mild (4–7 days), moderate (8–28 days), severe (+28 days).

**Table 4 ijerph-20-01121-t004:** Descriptive statistics and Wilcoxon signed rank test for GPS metrics in weekly performances (*n* = 23).

			Wilcoxon Signed Rank Test		
	Mean ± Standard Deviation		*z*	*p*	*r*
Variable	Before injury (–4W)	After injury (+4W)			
Work time (min)	482.2 ± 113.8	377.4 ± 61.1	–3.772	0.56	<0.01 *
TD (m)	32203.8 ± 8102.2	23466.5 ± 5059.9	–3.802	0.56	<0.01 *
HSR (m)	2218.4 ± 416.5	1591.9 ± 364.2	–4.015	0.59	<0.01 *
ACC (m)	290.2 ± 60.3	219.6 ± 43.8	–3.711	0.55	<0.01 *
DEC (m)	267.8 ± 74.1	204.1 ± 51.9	–3.346	0.49	<0.01 *
	Before injury (–4W)	Season average (S)			
Work time (min)	482.2 ± 113.8	338.1 ± 53.7	–3.772	0.56	<0.01 *
TD (m)	32203.8 ± 8102.2	21475.5 ± 3737.3	–3.893	0.57	<0.01 *
HSR (m)	2218.4 ± 416.5	1545.1 ± 233.1	–4.015	0.59	<0.01 *
ACC (m)	290.2 ± 60.3	203.9 ± 48.5	–3.802	0.56	<0.01 *
DEC (m)	267.8 ± 74.1	194.5 ± 56.6	–3.741	0.55	<0.01 *
	After injury (+4W)	Season average (S)			
Work time (min)	377.4 ± 61.1	338.1 ± 53.7	–1.460	0.22	0.144
TD (m)	23466.5 ± 5059.9	21475.5 ± 3737.3	–1.125	0.17	0.260
HSR (m)	1591.9 ± 364.2	1545.1 ± 233.1	–0.730	0.11	0.465
ACC (m)	219.6 ± 43.8	203.9 ± 48.5	–0.791	0.12	0.429
DEC (m)	204.1 ± 51.9	194.5 ± 56.6	–0.304	0.04	0.761

* *p* ≤ 0.01.

## Data Availability

The data presented in this study are available upon request from the corresponding author.

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
