# Peer review of "Weekly External Load Performance Effects on Sports Injuries of Male Professional Football Players"

_ijerph, 2023, doi:10.3390/ijerph20021121_

Round 1

Reviewer 1 Report

Summary: The authors provide an observational study on the influence of external load parameters (GPS) on the frequency of injuries within one football season for one professional football team. The novelty of this study results from a correlation between total weekly external load performance (training AND competition) and frequency of injuries. The authors present valuable results, showing higher external load performance levels 4 weeks before injury. These information can be used for adapting training loads and a possible value of regular GPS monitoring. Besides, the number of subjects examined is very limited, and data regarding type, frequency and location of injury as well as off-times must be interpreted with caution.

Abstract: ok.

Introduction: too long, limit references.

M&M: ok.

Results: ok.

Discussion: Focus on results regarding external load parameters and influence on injury frequency, less on type of injury and off-time (only one team in one season).

Individual comments:

52: what kind of study? please give some more information

73-74: what do you deduct from this? delete sentence

82: here you can add abbreviation TD

89-91: not clear enough, was the novelty of this investigation is, you should point that out in other words

98: prospective?

104-105: please give ranges

115: who is the medical department? team physician, physical therapist? how many?

126-127: isn't this the main difference compared to the existing literature?

135-137: please explain the choice of these time periods (4 weeks before and after)

150-154: please give standard deviations, and total days missed for the whole team due to all injuries

151: PER analyzed season?

157: not recurrent, but frequent

158-159: and table 3is there evidence for this differentiation between mild, moderate and severe? please give a reference

160-162: sentence makes no sense: list injuries clearly

187-189: indicate at this point, why the study is unique or offers novel information

204-205: please discuss a possible explanation for this difference

214: in the examined collective. why did the players have more muscle injuries?

270: it is

271: because

278ff limitations: small collective regarding the report of overall seasonal injuries (only one team, injury variations possible); investigation during corona pandemic, results may not be representative

Reviewer 2 Report

Thank you for the opportunity to review this manuscript entitled ‘Effects of Sports Injuries on Weekly External Load Performances of Male Professional Football Players.’  Although there are some interesting findings within the work and reporting of this nature are important.  I do feel the manuscript requires significant improvements.  My main issues are surrounding the study design, justification of the periods selected for analysis and ultimately the conclusions the authors have drawn because of the analysis completed.  In its current form I do not feel that it adds to the current body of literature in this area.  The authors would need to review the methodological design and resultant analysis of data to make it appropriate for publication. 

Abstract

This will require review in line with the comments made below.

Introduction

Line 55: Decreased roughly?  Feels like the sentence is incomplete

Line 55-56: Can we prevent injury or reduce risk? This is a topical debate in research currently, I would suggest rephrasing this. 

Line 57-63: I think this is a big paragraph to basically say that injuries are predominantly lower limb and that there is variation across countries with regards days absent from injury.  I would ask that any information is related to the population you tested, and the competition players partake in.  You have 33 players in your study are they all from the same league – Portugal?    Obviously, competition demand can influence injury.  Thus, any comparatives made in the intro must relate to your population. 

Line 66: What does training load mean?  Would be better to utilise language that represents your data collection, external demand, high velocity actions etc. 

Line 73-74: I would argue this is one thing we need to consider.  What about internal load, physiological response to demand, readiness to retrain? Re-word to define that this is one important consideration, but not the only consideration. 

Line 78-83: Language used here becomes descriptive and not critical.  In a study among, another study….  This becomes repetitive and demonstrates a lack of critique of existing literature…  Here for example, provide examples of the injuries, discuss the metric of TD – if this has increased it is likely that other metrics have come up, has this paper missed this in the discussion??  I don’t know because I have not read it, but critically evaluate the findings, to strengthen the need for your work. 

Line 80-83: We would expect this – this would be a ‘working day’ as highlighted by coaches, where game demand may be replicated.  This is also a 3-day pre game snapshot.  What does this add to the critical justification for your work?

Line 84-86: Please rephrase.  Analysis of games and training is extensive throughout literature. 

Methods

2.3 – Injury Report: How were players diagnosed? Who by and how were the injuries classified?  Important to note that if the physiotherapists diagnose then to acknowledge the variation that may occur between physios with regards classification.  Other considerations include: How experienced were the medical staff?  Was it a signal member of the medical team that diagnosed the injury?

Table 1: Deceleration is a negative value so >-3m.s2

Was SD not analysed? There would be a benefit of breaking down the HSR.  Greater than 5m/s would display a huge variation between this value and a sprint.  Particularly when analysing muscle injuries.  Recent data released is showing that muscle injuries are occurring at relatively low speed thresholds. By doing this you could run an analysis at what intensities the injuries were sustained which would add greater impact for your work. 

Line 136: revise sentence

What is the value of comparing to the season average for that player?  This average would be distorted due to the period out with injury and thus not reflect a full season.  Please justify.

Line 142: What do you mean by zone?

Line 143: This is a fundamental flaw of the work.  You discuss in the intro the need for individualisation of players physical metrics.  However, utilise absolute values for analysis.  We know that when we review absolute values of players, quicker players would report greater HSR metres, but does this represent high speed for them?  Also, by analysing absolute variables, we do not achieve a true reflection of when the non-contact injury occurred, as we would be unsure if ‘for that player’ they were achieving high velocity stimulus. 

Statistical analysis: the study would benefit from meaningful change analysis.  Also, could a Bland-Altman analysis be useful here to determine effect of the 2 4-week periods? I am not suggesting this is a necessity, but may be worth exploring to strengthen the findings/conclusions drawn.

Results

Table 2: Doesn’t represent the UEFA injury classification model of hrs missed.  This would then make the data comparable to injury audit research. 

Methods need to include how the injuries were classified – i.e. mild, moderate, severe….

Table 3: Injuries need to be classified in relation to exact location – i.e. hamstring, severity through grading (i.e. 3c).  Please review the work of Hagglund, Ekstrand and Walden over the last 20 years, this will detail how to classify injuries, and analyse in relation to time lost etc. 

Line 164-168: So, players returning 4 weeks was reduced post injury.  There are a number of factors you do not account for.  Were sessions adapted in this period for the returning players, were they involved in game, etc.  The 105 mins reduction could be a result of not playing?  As a part of returning to play I would normally expect a period of management and integration into training and game play for the player.  This would align to Taberner research in RTT and RTP protocols.  This would then make the differences pre injury to return less impactful as a finding.  With this in mind the rolling averages (4 weeks) until point of injury would be more useful.  I.e. was the 4 weeks pre injury considered a significant spike compared to the rolling averages pre this 4 week block. 

Again, I would expect season averages to reduce due to the period of injury where no GPS data was collected.  This does not represent an impactful or novel finding for me. 

Discussion/Conclusion

Due to the queries regarding the methodological approach and analysis completed, I do not think the authors have been able to address the aims of the paper and thus need to carefully consider the queries raised before any discussion or conclusions are drawn. 

Before a discussion or conclusion is reviewed the authors need to address the concerns highlighted above. 
